# The Roles of RNA-Binding Proteins in Vasculogenic Mimicry Regulation in Glioblastoma

**DOI:** 10.3390/ijms26167976

**Published:** 2025-08-18

**Authors:** Pok Kong Tsoi, Xian Liu, Man Ding Wong, Liang-Ting Lin

**Affiliations:** Department of Health Technology and Informatics, The Hong Kong Polytechnic University, Kowloon, Hong Kong, China; pok-kong.tsoi@connect.polyu.hk (P.K.T.); xian.liu@connect.polyu.hk (X.L.); man-ding-winson.wong@connect.polyu.hk (M.D.W.)

**Keywords:** RNA-binding proteins, glioblastoma, vasculogenic mimicry

## Abstract

Glioblastoma (GBM) is a highly aggressive brain tumour characterised by a poor prognosis and resistance to anti-angiogenic treatments. Vasculogenic mimicry (VM), in which tumour cells form vessel-like structures independent of endothelial cells, has emerged as a key mechanism hindering the efficacy of anti-angiogenic therapies. Recent research highlights the central role of RNA-binding proteins (RBPs) in regulating VM through diverse post-transcriptional mechanisms, including mRNA decay induction and translational repression. Several oncogenic RBPs, such as HuR and HNRNPs, promote VM and tumour aggressiveness, while others, including RBMS3, act as suppressors of VM. Despite the prominent oncogenic roles of multiple RBPs, RBP-targeting compounds aimed at suppressing VM in GBM have remained at an early stage due to a number of limitations. This review summarises the role of VM in the treatment resistance of GBM, RBP regulation of VM, and the current landscape and future direction of RBP-targeted therapies aimed at overcoming VM-mediated treatment resistance in GBM.

## 1. Introduction

Glioblastoma (GBM) is the deadliest and most common form of malignant glioma [1]. GBM treatment typically involves maximal surgical resection, followed by radiotherapy and chemotherapy. Despite the use of combination therapy, the median survival rate of GBM patients remains at 14.6 months, while the 5-year survival rate is only 5% [2,3]. Because of its dense and aberrant vasculature, agents targeting angiogenesis have been widely explored as a potential GBM treatment [4,5,6]. Bevacizumab, initially one of the most promising anti-angiogenic agents targeting vascular endothelial growth factor (VEGF), was approved by the Food and Drug Administration (FDA) after a successful phase II clinical trial. Bevacizumab prolonged overall survival (OS) to 19.6 months and progression-free survival (PFS) to 13.6 months in newly diagnosed GBM patients [7]. However, bevacizumab demonstrated no significant OS benefit in phase III clinical trials and is instead used for recurrent GBM as a second-line treatment [8]. It was later discovered that GBM cells undergo alternative forms of vascularisation, particularly vascular mimicry (VM), that escape the effects of VEGF inhibitors [9,10]. Since its initial discovery in 1999, VM has garnered considerable attention due to its initial controversy [11]. However, years of scientific debate have now established VM as a critical neovascularisation pathway in GBM, counteracting various anti-angiogenic agents.

VM refers to the formation of perfusion-capable structures formed by cancer cells independent of endothelial cells. Two main types of VM have been identified: a patterned matrix type and a tubular type. The patterned matrix type describes the formation of a looping matrix without a continuous lumen, yet surprisingly, capable of conducting plasma and blood cells [12]. The tubular type describes the de novo formation of tubular, vessel-like structures by glioblastoma stem-like cells expressing endothelial-associated genes [10]. Although considered the dominant blood supply in the early stages of some cancers, VM in GBM often acts as a resistance mechanism to anti-VEGF therapy, allowing for the neovascularisation of GBM despite anti-angiogenic treatment [13].

VM regulation depends on a combination of multiple carcinogenic pathways. These pathways involve the interplay between various regulatory elements, including transcription factors, RNAs, and RNA-binding proteins (RBPs). RBPs are key post-transcriptional regulators containing RNA-binding domains that control gene expression by modulating RNA activity. RNA-binding domains exist in different forms, such as RNA recognition motifs (RRMs), K-homology domains, and domains containing Arg-Gly-Gly repeats [14]. These domains enable RBPs to target a wide range of RNA sequences and structures, thereby regulating RNA splicing, transport, stability, translation, and other processes. Dysregulation of RBPs commonly leads to carcinogenesis and tumour progression. For example, human antigen R (HuR) and Lin28 are commonly upregulated oncogenic RBPs shown to promote GBM proliferation, invasion, angiogenesis, and immune evasion [15,16]. Given their regulatory significance, multiple RBPs have been considered prognostic markers for GBM, with significant efforts underway to use them as targets for GBM treatment.

Recently, an increasing number of RBPs have been discovered to regulate VM in GBM; however, a comprehensive understanding of their roles has yet to be established. Previous reviews have focused on RBPs in brain tumours in general as well as RNA modifications in GBM [17,18], but there is currently no in-depth review focused on RBPs in the VM regulation of GBM. In this review, we summarise the effects of VM on GBM treatment resistance, RBP regulation of VM in GBM, and the potential future directions for targeting different RBPs involved in the VM in GBM.

## 2. Vasculogenic Mimicry in Glioblastoma Treatment Resistance

### 2.1. Anti-Angiogenic Therapy Resistance

Anti-angiogenic therapy (AAT) was first proposed by Judah Folkman in 1971 when he discovered that neovascularisation was indispensable for tumour growth. Since then, the conventional focus of angiogenesis treatment has been to eradicate the vasculature in tumours, depriving cancer cells of nutrients and oxygen necessary for growth [19]. However, the anti-angiogenic agent bevacizumab demonstrated no significant efficacy in GBM, despite it being one of the most vascularised tumours. To explain the failure of AAT, R. K. Jain [20] proposed in 2001 that the goal of AAT should be to normalise the tumour vasculature by balancing proangiogenic and anti-angiogenic factors. Termed ‘vascular normalisation’, improving vascular function during a normalisation time window should reduce tumour hypoxia while enhancing the delivery of cytotoxic drugs.

However, current ‘vascular normalisation’ treatments for GBM have required further optimisation. While there are reports that normalisation helped to reduce oedema and increase PFS [21], there are mixed reports in phase II clinical trials on whether ‘vascular normalisation’ translates to OS benefits in GBM patients [22,23,24]. Some critics also suggest that, despite morphological normalisation, the function of vessels post-AAT remained unnormalised and that hypoxia increased with lower vessel permeability [25]. Furthermore, alternative neovascularisation pathways, particularly VM, could nullify the therapeutic effects of conventional AATs.

VM in GBM is characterised by leaky and disordered vessels. Although perfusion is possible within these vessels, the actual efficiency of vascular function is questionable. The incidence of VM in GBM ranges from 30 to 67%, and patients with VM-positive GBM generally have a shorter survival and poorer prognosis than those with VM-negative GBM [26,27,28,29,30,31,32,33,34]. Despite the high incidence of VM in GBM, a report by Maddison et al. [27] suggested that VM contributed to an insignificant amount of vasculature in GBM, questioning its importance. However, the study by Maddison et al. [27] only involved tumour samples from patients who underwent conventional therapy (surgery followed by chemotherapy and radiotherapy), and not patients who received AAT. While VM may not be the primary source of GBM angiogenesis, VM could be a significant factor in the resistance to AATs. A report demonstrated that six days after bevacizumab treatment in orthotopic GBM mouse models, the levels of vessel co-option, intussusceptive microvascular growth, and sprouting angiogenesis all decreased, while the level of VM increased significantly [35]. Another report using orthotopic GBM mouse models also showed that the number of VM structures increased after vatalanib treatment [36]. These reports suggested that VM could be the primary source of resistance to various anti-angiogenic agents.

Taken altogether, VM may be critical to the success of AAT. If VM is a form of abnormal neovascularisation contributing to poor perfusion, VM structures may need to be targeted to achieve the goal of ‘vascular normalisation,’ thereby prolonging the drug delivery window and reducing hypoxia-induced invasiveness. If current AATs, such as bevacizumab or vatalanib, fail to normalise the VM vessels, alternative targets may need to be pursued to improve the efficacy of AATs.

### 2.2. Radiotherapy and Chemotherapy Resistance

In addition to AAT resistance, VM is also implicated in radiotherapy and chemotherapy resistance (Figure 1). VM in gliomas is promoted by a transcription factor Prrx1, the levels of which were shown to correlate with a poorer response to chemotherapy [37]. Due to the aberrant nature of VM-related vessels with poor perfusion, poor temozolomide delivery to cancer cells is implied [37]. Furthermore, VM vessels were shown to be resistant to radiotherapy. A study by Lin et al. [38] demonstrated in vivo that no significant damage was incurred by VM-based networks after 6 Gy of irradiation compared to vessels formed by endothelial cells, suggesting that cells participating in VM-based networks may potentially evade radiotherapy and thus contribute to the recurrence of GBM.

The involvement of VM in combination therapy resistance highlights the importance of targeting VM in GBM. However, current reports are limited, and considering the constraints of in vitro studies, further research is recommended to validate the causal relationship between VM and treatment resistance on a larger scale.

## 3. Mechanisms of Glioblastoma Vasculogenic Mimicry

The molecular mechanism of VM involves a complex interplay between various pathways. These pathways include the Notch/NF-κB pathway [39], the tumour-associated macrophage-activated PGE_2_/EP_1_/PKC pathway involving COX-2 [40], and the MAPK/ERK pathways [41]. In the following subsections, the main pathways leading to VM in GBM, such as the phosphatidylinositol 3-kinase (PI3K)/Akt pathway, the matrix metalloproteinases (MMPs)/laminin 5γ2 (LAMC2) pathway, the vascular endothelial cadherin (VE-cadherin) pathway, and the transforming growth factor β (TGF-β) pathway, are summarised.

### 3.1. The PI3K/Akt Pathway

The PI3K/Akt pathway is a commonly upregulated pathway key to VM formation in GBM [42]. The activation of the PI3K/Akt pathway is mainly governed by upstream receptor tyrosine kinases (RTKs) [43]. When RTKs bind to their respective substrates, PI3K is phosphorylated. Phosphorylated PI3K then phosphorylates PIP2 to PIP3, which activates Akt. Activated Akt then regulates several downstream effectors, commonly mTOR, FOXO, and GSK3, which control different malignant characteristics of GBM, such as apoptosis evasion, immune escape, and chemotherapy resistance [43,44,45]. Conversely, suppression of the PI3K/Akt pathway could inhibit several key hallmarks of GBM, such as cell survival, proliferation, and migration, making PI3K/Akt a prime target for GBM treatment [46]. Some of the upstream regulators of the PI3K/Akt pathway leading to VM include vascular endothelial growth factor receptor 2 (VEGFR-2), epidermal growth factor receptor (EGFR), ephrin type-A receptor 2 (EphA2), and L1 cell adhesion molecule (L1CAM).

VEGFs and their receptors, VEGFRs, are one of the most common RTKs involved in conventional GBM angiogenesis. In GBM VM, VEGFR activation is different from that of conventional angiogenesis in two ways: First, VEGFR-2, but not VEGFR-1, seems to be responsible for VM formation through glioma stem-like cells [47]. Second, VEGFR-2 is activated by reactive oxidative species in addition to VEGF [41]. Once phosphorylated on tyrosine residue Y1175, VEGFR-2 activated PI3K/Akt pathways in U87 cells, leading to VM formation both in vitro and in vivo [42,47].

EphA2 is another RTK that commonly promotes VM in GBM. EphA2 is phosphorylated by PLK4 at two sites, Ser891 and Ser901 [48]. Activated EphA2 was then shown to activate the PI3K pathway, which promoted the VM formation, migration, and proliferation of glioma cells [48]. Conversely, the knockdown of *EPHA2* in glioma stem cells downregulated extracellular matrix (ECM)-related protein expression of MMPs, which inhibited VM formation in vitro and in vivo [49].

EGFR is also an upregulated RTK that promotes the VM in GBM under hypoxic conditions. An antagonist of EGFR, LRIG1, suppressed the expression of phosphorylated EGFR when overexpressed under hypoxia. Thus, VM formation, proliferation, migration, and invasion is reduced by deactivating the PI3K/Akt pathway [50]. As a tumour suppressor, downregulating LRIG1 also led to a more aggressive tumour type [51].

Finally, L1CAM is also recognised to regulate VM formation in GBM by upregulating various VM-associated proteins. L1CAM is upregulated in glioma and was associated with a poor prognosis in patients [52]. Overexpressed L1CAM was shown to downregulate the microRNA miR-143-3p, which reduced its binding to the *HK2* gene [52]. The resulting increased expression of HK2 upregulated the PI3K/Akt pathway, increasing the expression of MMPs and VEGF-A, which induced VM formation [52].

### 3.2. The VE-Cadherin Pathway

VE-cadherin, also known as CDH5 or CD144, is another key protein commonly upregulated VM-related cells in GBM, particularly in glioma stem-like cells, where it facilitates tumour vascularisation and microenvironment remodelling. In an experiment involving glioma stem-like cells, HIF-1α and HIF-2α bound to the promoter of VE-cadherin to upregulate its expression [53]. The increased VE-cadherin expression contributed to a two-fold increase in VM formation by glioma stem-like cells, an effect that was further enhanced under hypoxia [53]. VE-cadherin expression was also attenuated by *KDR* knockdown in glioma stem-like cells [47]. Furthermore, VE-cadherin was shown to interact with other pathways, such as the MAPK/ERK, PI3K/Akt, and MMP/laminin pathways, to promote VM [54,55]. Therefore, VE-cadherin has been accepted as a VM-related marker along with MMPs.

### 3.3. The TGF-β Pathway

TGF-β is a cytokine that regulates various GBM processes, including proliferation, invasion, immunosuppression, and stemness. Recently, TGF-β was also found to contribute to VM in GBM. TGF-β binds to TGF-β receptor II and phosphorylates TGF-β receptor I, thereby triggering both Smad-dependent and non-Smad pathways [56]. TGF-β is also known to regulate conventional angiogenesis by increasing the expression of IGFBP7 and VEGF [57,58]. Recently, TGF-β was found to bind β8 integrin and promote VM formation by regulating Smad phosphorylation and RhoA signalling [59]. Upon β8 integrin (ITGB8) knockdown, VM and epithelial–mesenchymal transitions (EMT)-related proteins, such as VE-cadherin, MMP2, and N-cadherin, were also downregulated [59]. Furthermore, downregulation of TGF-β was also found to reduce MMP expression and subsequently inhibit tubule formation of U251 cells [60].

### 3.4. Hippo/YAP Pathway

The Hippo/YAP pathway is another pathway contributing to GBM VM. Recently, the transcripts of YAP1 and TEAD1, both critical components of the Hippo signalling chain, were found to be upregulated in GBM tissue [61]. YAP1 and TEAD1 upregulation correlated with increased VM levels in several GBM cell lines, and knocking down these proteins by using their corresponding inhibitors successfully reduced VM levels in vitro [62]. Therefore, YAP signalling pathway activation may contribute to VM formation by increasing levels of the downstream targets of Snail and FOXC2 [62].

### 3.5. Matrix Metalloproteinases and Laminin Pathway

Of the pathways described above, the ultimate executor of VM seems to involve ECM remodelling. Thus, MMPs and laminin chains come into play. As a critical component of the ECM, laminin forms the basement membrane and regulates cell adhesion, migration, and invasion [63]. MMPs cleave laminins. Specifically, MMP14 has been shown to convert proMMP2 into MMP2, which is then involved in the cleavage of laminin proteins into different subunits [60]. Fragmented laminins then participate in VM formation. To date, MMPs and laminins are among the most extensively examined markers of VM.

### 3.6. RNAs in Glioblastoma Vasculogenic Mimicry Formation

Recently, RNAs have garnered more attention for their role in VM regulation. There are increasing more studies reporting how different types of RNA, such as microRNA, circular RNA (circRNA), and long noncoding RNA (lncRNA), govern the expression of different VM-related proteins [64,65,66]. For example, LINC00339 overexpression in glioma cells suppressed miR-539-5p expression, which promoted TWIST-1, MMP2, and MMP14, leading to an increase of VM in vitro and in vivo [67]. By modulating the expression of LINC00339 and miR-539-5p, VM formation was suppressed both in vitro and in vivo, lengthening the survival of xenograft mice [67].

## 4. RNA-Binding Proteins in Vasculogenic Mimicry in Glioblastoma

As governors of RNA functions, RBPs also participate in VM formation. The RBPs involved in VM can be classified into mRNA-binding proteins, circRNA-binding pro-teins, lncRNA-binding proteins, and N6-methyladenosine (m6A) modulation proteins. Table 1 summarises the known RBPs controlling GBM VM.

### 4.1. Messenger RNA-Binding Proteins

Most mRNA-binding proteins regulate the stability of mRNAs to control VM in GBM (Figure 2).

#### 4.1.1. HuR

HuR is a key oncogenic RBP in GBM that was recently found to control VM. It contains three RRMs that preferentially bind to adenine and uridine-rich elements (AREs) in the three-prime untranslated region (3′ UTR) of mRNAs. HuR was found to bind to *VEGF-A* mRNA, preventing its degradation [68]. Increased *VEGF-A* expression promoted phosphorylation of VEGFR2 and Akt, which promoted VM formation by increasing Snail, E-cadherin, and vimentin expression [68]. In addition to its role in VM, HuR is also known to bind to more than 30 other mRNA transcripts, including *BCL2*, *MYC*, and *IL10*, corresponding to different cancer hallmarks such as evasion of apoptosis, proliferation, and immune suppression [15]. The multifaceted roles of HuR thus render it a compelling therapeutic target.

#### 4.1.2. Pumilio Homolog 2 (PUM2)

PUM2 was recently discovered to be a suppressor of VM in GBM. However, it is uncertain whether PUM2 functions as a tumour suppressor in GBM. PUM2 is a member of the PUF protein family, which recognises the pumilio homology domain and acts as a translational repressor by binding to the 3′UTR [83]. Some researchers suggested that PUM2 levels become elevated and contribute to proliferation and migration by suppressing BTG1 expression [84], while others have suggested that PUM2 has low expression in GBM cells compared to normal brain tissue due to its SUMOylation by Ubc9 and subsequent degradation [69]. PUM2 degradation was found to increased *CEBPD* mRNA expression, which, in turn, promoted *DSG2* expression and enhanced the migration, invasion, and VM formation capabilities both in vitro and in vivo [69]. It is possible that the expression levels varied due to the intrinsic heterogeneity of GBM cells.

#### 4.1.3. Heterogeneous Nuclear Ribonucleoproteins (HNRNPs)

Three members of the HNRNP family have been shown to promote VM in GBM. First, HNRNP A2/B1 was found to increase the stability of *NFATC3* mRNA by binding to its 3′UTR. NFATC3, a transcription factor, worked synergistically with FOSL1 to enhance *VEGFR-2* expression, thereby driving VM, proliferation, and migration of GBM cells [70]. Another family member, HNRNP D, promoted the decay of tumour suppressor *ZHX2* transcripts, and regulated VM through the linc00707/miR-651-3p/SP2 axis, with SP2 directly regulating VM-associated proteins, including MMP2, MMP9, and VE-cadherin [71]. In an in vivo study, mice xenografts transplants with HNRNP D and with linc00707 knocked down, combined with ZHX2 overexpression, also showed the longest survival time, the smallest tumour volume, and the lowest density of VM structures [71]. Additionally, post-translational modifications on HNRNPs were also found to control VM, as shown by the SUMOylation of RALY. The increased SUMOylation of RALY at K175 by upregulated UBA2 stabilised *FOXD1* mRNA [72]. FOXD then bound to the promoter region of the *DKK1* gene, leading to the increased expression of MMP2, MMP9, and VE-cadherin [72]. HNRNPs mainly serve as nuclear m^6^A readers that recognise methylated RNA sequences [85]; however, there is currently no specific evidence linking their regulation of VM in GBM to their m6A-related functions. HNRNPs therefore promote VM in GBM by regulating the stability of mRNAs.

### 4.2. Circular RNA-Binding Proteins

The control of VM in GBM by circRNA-binding proteins involves regulating both the splicing of pre-mRNAs and the stability of mRNAs (Figure 3).

#### 4.2.1. RNA-Binding Motif Single-Stranded Interacting Protein 3 (RBMS3)

RBMS3, a tumour suppressor RBP belonging to the c-Myc family, also regulates VM. In GBM, RBMS3 downregulation was found to decrease the expression of circHECTD1 by binding to its flanking introns containing the sequence UAUAUA [73]. The decreased circHECTD1 expression then reduced the ubiquitin-mediated degradation of NR2F1 by reducing translation of a 463 amino acid peptide [73]. The reduced degradation of the transcription factor NR2F1 led to elevated expressions of several VM proteins, including MMP2, MMP9, and VE-cadherin, which contributed to VM formation in vitro and in vivo [73]. RBMS3 downregulation also has a central role in regulating the EMT and metastasis in other types of tumours, including breast and hepatocellular carcinoma [86,87]. However, there are only limited reports on other tumorigenic pathways that are regulated by RBMS3 in GBM.

#### 4.2.2. The Serine/Arginine-Rich Splicing Factor (SRSF) Family

Two members of the SRSF family, SRSF1 and SRSF7, were found to promote VM formation in GBM. The SRSF family is considered oncogenic in GBM, with members such as SRSF1, SRSF3, and SRSF9 promoting GBM proliferation, migration, and tumorigenesis [88,89,90,91]. SRSF1 contains two RRMs, with RRM1 preferentially recognising CN motifs and RRM2 recognising GGA motifs, both of which provided splicing control [92]. In VM, SRSF1 stabilised circCMTM3 upon its entry into differentiated glioma cells via exosomes, preventing its degradation [74]. circCMTM3 is then found to increase the expression of phosphorylated STAT5A; STAT5A subsequently bound to the *SRSF1* promoter region, further upregulating *SRSF1* expression [74]. This feedback loop between SRSF1, circCMTM3, and STAT5A promoted the VM formation, proliferation, invasion, and migration of glioma cells by upregulating CHI3L2 [74]. Another SRSF member, SRSF7, was also found to be upregulated in glioma [75]. SRSF7 increased the expression of circPLEKHA5, which encoded a 622 amino acid peptide that led to increased VM formation, migration, invasion, and proliferation through the enhanced expression of MMP2 and VE-cadherin [75].

### 4.3. Long Non-Coding RNA-Binding Proteins

VM regulation of GBM by lncRNA-binding proteins commonly involves controlling Staufen-mediated mRNA decay (SMD) and regulating lncRNA stability (Figure 4).

#### 4.3.1. Poly(A) Binding Protein Cytoplasmic 5 (PABPC5)

PABPC5 is a member of the poly A binding protein family localised to the cytoplasm, where it mediates RNA stability and transport. PABPC5 was found to promote VM formation of glioma cells by stabilising the lncRNA HCG15 [76]. HCG15 then promoted the decay of *ZNF331* mRNA by SMD [76]. Decreased ZNF331 protein levels, resulting from decreased *ZNF331* mRNA levels, then increased LAMC2 expression to contribute to VM in GBM [76]. Currently, there are only a few reports on the role of PABPC5 in GBM, while its sister protein, PABPC1, has garnered much attention. PABPC1 is a key driver of multiple cancer hallmarks in various cancers, such as angiogenesis and invasion [93]. In GBM, however, PABPC1 was reported to function as a tumour suppressor through its interaction with a lncRNA [94].

#### 4.3.2. T-Cell Intracellular Antigen 1-Related Protein (TIAR)

TIARs are proteins involved in stress granule formation that were recently found to suppress VM formation in GBM. They bind to AU-rich elements on RNAs to suppress the protein synthesis of several translation factors under cellular stress, such as the eIF family and c-Myc [95]. In GBM, reduced TIAR expression was reported to stabilise the lncRNA LOXL-AS1. LOXL-AS1 then downregulated miR-374b-5p, leading to increased MMP14 expression and VM formation in U87 and U251 cells [77].

#### 4.3.3. Zinc Finger Ran-Binding Domain-Containing Protein 2 (ZRANB2)

ZRANB2 is an alternative splicing regulator that is overexpressed in GBM and promotes VM. It contains two zinc finger domains that recognise AGGUAA motifs [96]. ZRANB2 upregulation stabilised the lncRNA SHNG20, increasing *FOXK1* mRNA degradation via the SMD pathway [78]. FOXK1 is a key transcriptional repressor of several VM-related proteins, including MMP1, MMP9, and VE-cadherin. Reduced FOXK1 expression promoted VM formation in U87 and U251 cells [78].

#### 4.3.4. Insulin-like Growth Factor 2 mRNA-Binding Protein 2 (IGF2BP2)

IGF2BP2 is a commonly upregulated cytoplasmic oncogenic RBP that may also regulate VM. When IGF2BP2 is SUMOylated at K497, K505, and K509 by SUMO1, the half-life of IGF2BP2 increases, which was found to upregulate lncRNA OIP5-AS1 [79]. OIP5-AS1 subsequently suppressed miR-495-3p, reducing its availability to bind to the 3′UTRs of *HIF1A* and *MMP14* mRNAs [79]. The consequent upregulation of HIF-1α and MMP14 promoted VM formation in vitro and in vivo [79]. While IGF2BP2 primarily functions as an m^6^A reader targeting methylated RNAs [94], there are no reports relating its m6A recognition properties to VM-promoting mechanisms.

### 4.4. Methyltransferase-like 3 (METTL3)

The role of METTL3, a m^6^A methyltransferase, is controversial (Figure 5). In one study with U87 and U251 cells, upregulated METTL3 in U87 and U251 cells stabilised the lncRNA HOTAIRM1 and contributed to VM formation by increasing IGFBP2 expression [81,97]. METTL3 was also found to be overexpressed in U373 cells, where it methylates and enhances the stability of BUD13 mRNA and contributes to VM formation via the BUD13/CDK12/MBNL1 axis [82]. On the contrary, a study with U87 cells showed that mRNA methylation levels decreased when METTL3 was downregulated [80]. The reduced mRNA methylation levels were found to promote both VM formation and the EMT of GBM cells, as indicated by the increased expression of MMP2, CDH2, and FN1 proteins [80]. These conflicting reports supporting METTL3 as both a tumour suppressive and oncogenic RBP could be due to the heterogeneous nature of GBM cells.

To summarise, RBPs regulate core VM processes through canonical and non-canonical VM pathways. RBPs such as HuR and HNRNP A2/B1 activate the VEGFR-2/PI3K/Akt pathway. The majority of RBPs, including HNRHP D, RALY, RBMS3, SRSF7, and ZRANB2, regulate VM through MMP/LAMC2 and VE-cadherin pathways. RBPs such as PABPC5, TIAR, and IGF2BP2 mainly control VM through the MMP pathway. PUM2 and SRSF1 modulate VM by regulating the non-canonical DSG-2 and CHI3L2 pathways, respectively. METTL3 dynamically participates in both the activation and inhibition of VM signals primarily through canonical MMP and VE-cadherin pathways. As “molecular switches” of pathways, RBPs have high potential to block VM in GBM.

## 5. Exploring the Possibilities of RNA-Binding Proteins as Therapeutic Targets in Vasculogenic Mimicry Formation of Glioblastoma

Despite the prominence of RBPs in cancer physiology, there are currently no FDA-approved compounds targeting RBPs in cancer treatment. Clinical trial information on seven RBP-targeting drugs can be found in a recent review by Wang et al. [18]. RBPs are notoriously difficult to target and were once considered undruggable. The challenges of targeting RBPs was previously reviewed [98,99]. Several key factors hinder RBP therapeutic design, including their intracellular localisation, the lack of hydrophobic binding pockets, and off-target toxicities. These factors may contribute to the scarcity of clinical trials on RBP inhibitors. Thus, drugs that target VM in GBM by targeting RBPs are even scarcer, with only one compound, juglone, discovered to inhibit VM in GBM by targeting HuR.

### 5.1. Juglone: A Novel Inhibitor of Glioblastoma Vasculogenic Mimicry Targeting HuR

HuR appeared to be a prioritised RBP target in GBM due to its well-established and ex-tensive oncogenic effects. Juglone is a natural small-molecule inhibitor of HuR, commonly found in walnut trees. It produces potent anticancer effects by inducing apoptosis, inhibiting angiogenesis, and preventing migration [100]. In GBM, juglone was shown to block RRM1/2 of HuR, thereby preventing HuR from binding to *VEGFA* mRNA. Thus, VEGF-A was downregulated, which reduced VEGFR-2 RTK activation and PI3K/Akt pathway deactivation, consequently suppressing the EMT and VM [68]. Juglone’s ability to cross the blood–brain barrier (BBB) was also confirmed in U251 orthotopic mouse models, in which intraperitoneal injections of juglone increased the survival rate of mice compared to controls [68]. In addition to inhibiting VM, juglone also induced apoptosis, ferroptosis, and ROS generation in GBM by inducing p38 MAPK phosphorylation [101,102]. However, the cytotoxic effects of juglone may not be limited to cancer cells. Juglone was also shown to induce DNA damage and inhibit transcription by degrading p53 in normal human fibroblasts [103]. Thus, juglone modifications or delivery methods should be carefully chosen to minimise the negative effects of juglone on normal tissues. Despite the potential risks, juglone may represent the first potent anti-VM and cytotoxic agent for GBM treatment.

### 5.2. Possible Strategies—Small Molecule Inhibitors, Decoy Oligonucleotides, and Post-Translational Modification Targeting

In addition to juglone, HuR is also targeted by several small-molecule inhibitors, including MS-444, SRI-42127, and DHTS. All three inhibitors primarily prevented HuR multimerisation by obstructing the RRM domains, which subsequently prevented HuR translocation to the cytoplasm [104,105,106,107]. Specifically, MS-444 demonstrated anti-angiogenic activity by reducing VEGF and HIF-1α expression [106].

Decoy oligonucleotides are another method of targeting RBPs. By building tandem repeats of nucleotides mimicking the RRMs of RBPs, the nucleotides act as RBP sponges that attract and bind to RBPs, thereby competitively blocking the normal functions of the RBPs. One example is the SRSF1 decoy featuring three repeats of the CGCAGGA motif. Preliminary investigations showed that the inhibition of SRSF1 by these oligo-nucleotides increased tumour apoptosis and reduced GBM growth in vivo [108].

In addition to directly targeting RBPs, targeting their post-translational modifications may be another means of modulating RBP activity. The TAK-981 small molecule inhibitor competitively binds to a SUMO-activating enzyme to prevent the SUMOylation of proteins [108]. By decreasing the SUMOylation of HNRNP A2/B1, TAK-981 suppressed GBM angiogenesis and proliferation both in vitro and in vivo [109]. Unfortunately, TAK-981 has poor BBB penetration, and further modifications are warranted prior to its use in clinical applications.

### 5.3. Translational Potential of Vasculogenic Mimicry Targeting

To further translate VM targeting by RBPs into clinical applications, lessons can be learned from CVM-1118, currently the only drug targeting VM that is being tested in a clinical setting. CVM-1118 (foslinanib) is a phosphoric ester targeting TRAP1, a chaperone in the mitochondria [110]. Recent phase IIa trials with CVM-1118 for advanced neuroendocrine cancer demonstrated a PFS of 6.9 months, higher than the previously reported PFS of 4–6 months, with an acceptable tolerance [111,112].

Given the initial success of the first VM-targeting drug for advanced cancers, two key points can be drawn regarding VM-related drug development in cancer:

The drug must be specific enough to target only the desired molecule.

The target should control a variety of molecular pathways for maximising efficacy and minimising compensatory mechanisms.

Regarding the first point, TRAP1 belongs to the heat shock family of proteins, which includes other similar proteins such as Hsp90. Chu et al. [110] demonstrated through docking analysis that CVM-1118 only interacted with TRAP1, and not with Hsp90, thereby reducing potential side effects due to off-targeting [110]. Second, targeting TRAP1 directly affected HIF-1α, which controls several downstream pathways that initiate VM in cancer cells, notably those of Nodal, VEGF-A, EphA2, and TWIST1 [110]. Further biomarker research found that the efficacy of CVM-1118 may be enhanced in patients with *NF2*- or *STK11*-deficient mutations, thereby solidifying the role of CVM-1118 in precision medicine [110].

Although TRAP1, the target of CVM-1118, is not an RBP, they are both intracellular proteins that regulate multiple downstream pathways. The initial success of VM inhibition through TRAP1 suggests that, with careful selection, RBPs may be promising targets for disrupting VM and suppressing cancer progression. However, BBB permeability remains a major obstacle for targeting VM in GBM, as many screened compounds have failed to achieve adequate penetration through the BBB, warranting new tools for more accurate predictions and rapid drug screening.

## 6. Perspectives

Although the prevalence of VM in GBM has been widely documented, the current understanding of VM detection, origins, and functional characterisations are insufficient. Through the advent of single-cell multi-omics (genomics, transcriptomics, and proteomics), spatial biology, and artificial intelligence, a multiplexed approach may support future studies in the areas of VM marker identification and functional profiling, RBP-RNA interactions in VM-related GBM cells, detailed genomic profiling of different GBM subtypes, and new therapeutic approaches targeting RBP-RNA interactions in GBM VM.

### 6.1. Vasculogenic Mimicry Marker Identification and Functional Profiling

The generally accepted method for identifying VM in tissue samples uses a combination of PAS+ and CD34− staining [113]. However, neither PAS nor CD34 are direct markers for identifying VM. PAS staining is mainly used to identify glycogen, while CD34 is a marker primarily for identifying stem cells or endothelial cells. Thus, a more specific marker or different supplemental markers are needed to more accurately identify VM [114]. This may possibly be achieved through a multiplexed approach, using imaging-based methods such as Xenium or Nanostring. These protocols can analyse single-cell RNA and protein expression levels in formalin-fixed, paraffin-embedded tissue samples without the loss of spatial information [115,116], which may help to more accurately identify VM-specific cellular markers based on spatial information.

In addition to marker identification, spatial information may also be helpful for further identifying relationship between VM and other components of the tumour microenvironment. Although current data suggest that the number of VM vessels increased after AAT of GBM, the proportion of VM vessels to total tumour vasculature following AAT has not been investigated. With spatial biology at the single-cell resolution, the ratio of structures may be more accurately counted at a cellular level, thus eliminating bias and errors when manually evaluating regional selections, which may clarify the true extent of the contribution of VM to resistance against combined treatment approaches.

Furthermore, machine learning could be applied to analyse multi-omics data and reveal connections between VM-type GBM cells and other cancer cell types across a wide range of samples, including patient-derived and commercially available cell lines. The integration of these analytical techniques and predictive models may guide future mechanistic studies to elucidate how VM contributes to combined therapy resistance.

### 6.2. In Situ Mass Profiling of RBP-RNA Interactions

Future studies at the molecular level could incorporate multidimensional datasets to produce more comprehensive profiling of RBP-RNA interactions at the single-cell level. Current crosslinking immunoprecipitation (CLIP) techniques, such as CLIP-seq [117], miCLIP [118], and Ribo-seq [119], allow direct RBP-RNA interactions to be determined, thus quantifying epitranscriptomic modifications and profiling translational dynamics. When combined with methods such as assay of reverse transcription-based RBP binding site sequencing (ARTR-seq) and deamination adjacent to RNA modification target sequencing (DART-seq) that allow RBP-RNA-binding sites to be characterised in situ [120,121], sufficient data may be generated for computational predictions of a regulatory map that combines splicing, stability, and translational efficiency of each RBP-RNA pair with high spatial resolution. Spatial resolution is critical in the context of VM as not all tumour cells are capable of VM formation. Ultimately, the functional significance of RBMS3 and PABPC5 in GBM may be established by CLIP-seq and ARTR-seq across different cell lines, while m^6^A-miCLIP and DART-seq may help elucidate how the m6A-recognition functions of IGF2BP2 and HNRNPs contribute to the regulation of VM.

### 6.3. Accounting for Genomic Variations Across Glioblastoma Subtypes

Further research is needed to explore the effects of different GBM subtypes on the incidence of VM at the genomic level. GBM can be classified into four molecular sub-types based on gene expression and mutation profiles: classical, mesenchymal, proneural, and neural. Classical GBM is characterised by *EGFR* amplification or mutations together with *TP53* mutations [122], while mesenchymal GBM is characterised by NF1 inactivation, which abolishes the GTPase-activating function of neurofibromin, resulting in constitutive Ras-GTP loading [123]. EGFR and Ras are related to the PI3K/Akt/mTOR and MAPK pathways, respectively, that may regulate VM in GBM. Furthermore, the proneural type of GBM, defined by mutations in isocitrate dehydrogenase (IDH) and the accumulation of oncometabolite 2-hydroxyglutarate (2-HG), shows increased expression of several VM-related proteins such as VE-cadherin, HIF-1α, and EphA2 [124]. However, the relationship between the neural subtype and VM remains unreported. Thus, future research may also identify differences in VM levels of GBM subtypes.

The conflicting functions of RBPs in GBM may also be attributed to genomic variations in different subtypes. Wang et al. [125], using datasets from The Cancer Genome Atlas and the Chinese Glioma Genome Atlas, observed that RBPs were significantly downregulated in the neural subtype relative to the classical, mesenchymal, and proneural subtypes, and this downregulation was independent of tumour grade. These differences may explain the conflicting roles of PUM2 and METTL3 reported in various GBM studies. Furthermore, the proneural type of GBM may influence m6A methylation levels in GBM cells [126]. Crosstalk between 2-HG and m6A regulatory proteins may further complicate the elucidation of METTL3 effects in different GBM subtypes. Thus, future research in GBM genomics may focus on confirming the expression levels and functional roles of PUM2 and METTL3 in various GBM subtypes, for a more comprehensive picture of RBP regulation in GBM.

### 6.4. Blood–Brain Barrier Penetration and Target-Based Drug Discovery

Identifying drugs that can penetrate the BBB remains a major challenge, as the tight junctions formed by endothelial cells, pericytes, and astrocytes limits drug permeability. Traditionally, passive diffusion-based BBB-penetrating drugs have been selected using the central nervous system multiparameter optimisation (CNS MPO) system according to scores of six physiochemical parameters [127]. However, recent machine learning approaches have been able to increase the BBB-penetrating prediction accuracy to greater than 90% [128], which may significantly accelerate the drug discovery process for small molecules targeting RBPs. In addition to small molecules, future delivery strategies may use transporters. Jin et al. [129] developed tHFn(+) nanocarriers that encapsulated small-interfering RNA (siRNA) to pass though transferrin protein channels of the BBB. These siRNAs, containing specific RRMs, may represent another means of targeting specific RBPs in the brain.

Furthermore, AI may help to streamline drug discovery pipelines. First, deep learning analysis of existing RBP-networks may help to prioritise candidate RBP targets based on network centrality and associations with VM-related phenotypes. These models may help to predict regulatory networks and elucidate key axes in GBM initiation and progression [130,131]. Validating findings from in vitro assays (e.g., CRISPR-based RBP knockout in patient-derived cells) and in vivo models (orthotopic xenografts and organoids) [132] may help to confirm the functional relevance of core regulators that drive VM and GBM treatment resistance. These tools may facilitate further screening of existing RBP-targeting agents, such as MS-444, SRI-42127, and DHTS, for their VM suppressing abilities. The pharmacokinetics and BBB-penetrating ability of SRSF1-targeting oligonucleotides may also be better predicted by utilising AI. Looking ahead, research efforts could first prioritise the validation of existing RBP-targeting compounds before embarking on the discovery of novel RBP inhibitors. Alongside RBP-targeting, the heparanase inhibitor RDS3337 may further enhance efficacy against VM by targeting VM-associated pathways such as MMP and PI3K/Akt pathways, and serve as a complementary method for VM suppression in GBM [133,134]. Ultimately, the integration of high-throughput drug screening, nanodelivery technologies, and AI-assisted molecular design may accelerate and refine target-specific drug screening, potentially saving time and resources required for clinical translation.

## 7. Concluding Remarks

AAT in GBM has remained stagnant for the past 20 years, despite various efforts to improve its efficacy. Originally, sprouting angiogenesis was thought to be the sole form of angiogenesis; however, it is now clear that alternative neovascularisation pathways exist in GBM, rendering traditional single-target VEGF inhibitors ineffective. VM, as an essential neovascularisation pathway contributing to combined treatment resistance in GBM, may be regulated by RBPs; RBPs are therefore potential therapeutic targets in GBM. Currently, 13 RBPs have been identified that regulate VM in GBM, yet there are no FDA-approved RBP-targeting drugs. With the advent of AI, single-cell multi-omics, and spatial biology, the historically impossible target, RBP, may finally be targeted, potentially leading to new breakthroughs in AAT that can prolong GBM patient survival.

## Figures and Tables

**Figure 1 ijms-26-07976-f001:**
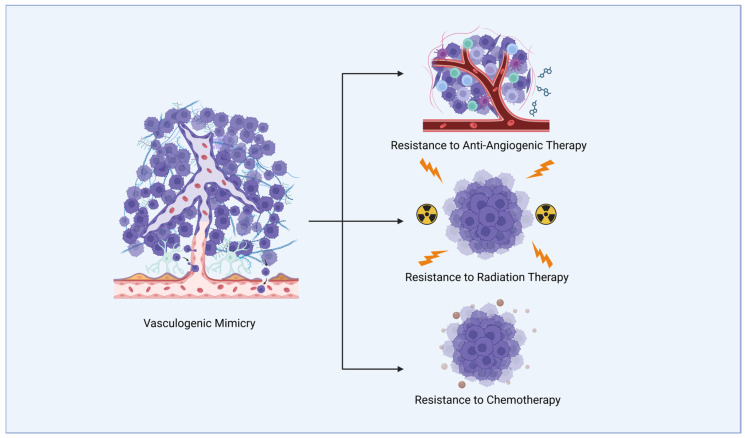
Vasculogenic mimicry (VM) refers to vessel-like structures formed by glioblastoma (GBM) cells (represented in purple) that can conduct blood cells. VM is implicated in resistance to different therapeutic modalities in glioblastoma (GBM) treatment.

**Figure 2 ijms-26-07976-f002:**
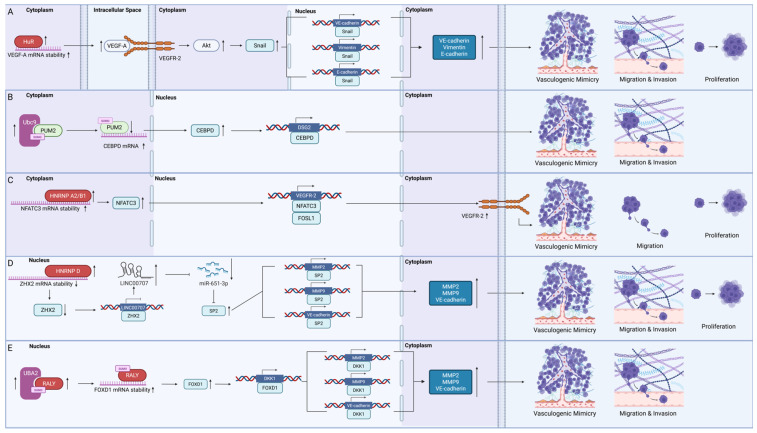
Messenger RNA-binding proteins (RBPs) involved in GBM VM. RBPs that promote VM are shown in red, while those that inhibit VM are shown in green. Transcription factors appear in light blue, proteins involved in SUMOylation are depicted in purple, and other pathway proteins are coloured very light blue. (**A**) VM regulation by human antigen R (HuR). (**B**) VM regulation by Pumilio homolog 2 (PUM2). (**C**) VM regulation by heterogeneous nuclear ribonucleoprotein A2/B1 (HNRNP A2/B1). (**D**) VM regulation by heterogeneous nuclear ribonucleoprotein D (HNRNP D). (**E**) VM regulation by RALY heterogeneous nuclear ribonucleoprotein (RALY) (created in BioRender).

**Figure 3 ijms-26-07976-f003:**
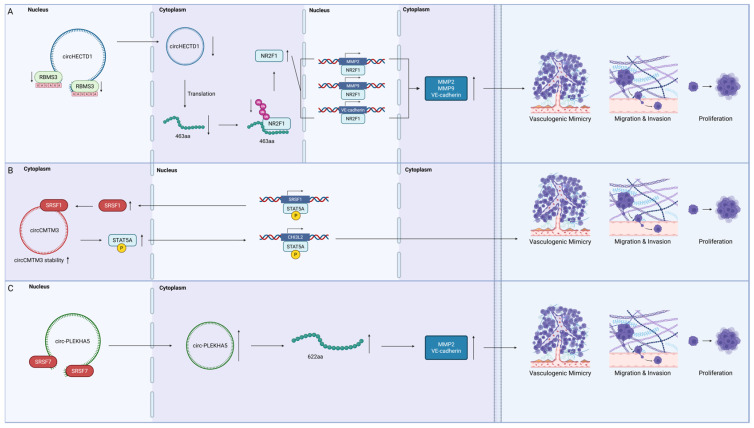
Circular RNA-binding proteins involved in VM of GBM. RBPs that promote VM are shown in red, while those that inhibit VM are shown in green. Transcription factors are coloured light blue; phosphorylation and ubiquitination are coloured yellow and pink, respectively; and amino acid chains are depicted in green. (**A**) VM regulation by RNA-binding motif single-stranded interacting protein 3 (RBMS3). (**B**) VM regulation by serine/arginine-rich splicing factor 1 (SRSF 1). (**C**) VM regulation by serine/arginine-rich splicing factor 7 (SRSF 7) (created in BioRender).

**Figure 4 ijms-26-07976-f004:**
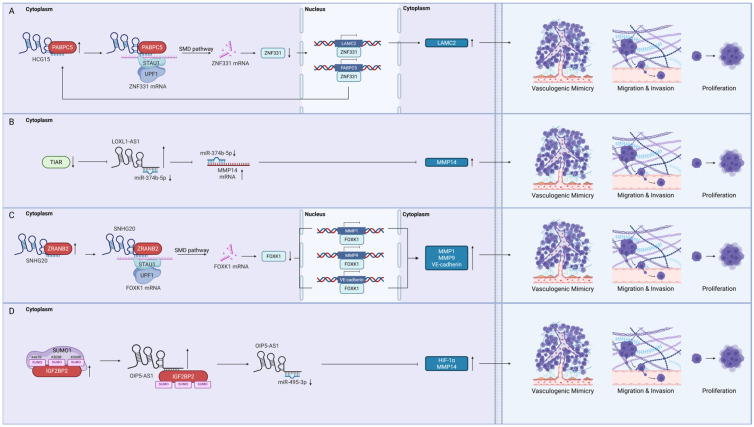
Long non-coding RNA-binding proteins involved in GBM VM. RBPs that promote VM are shown in red, while those that inhibit VM are shown in green. Transcription factors are coloured light blue, and Staufen-mediated decay proteins STAU1 and UPF1 are coloured medium blue and deep blue, respectively. (**A**) VM regulation by poly(A) binding protein cytoplasmic 5 (PABPC5) (**B**) VM regulation by T-cell intracellular antigen 1-related protein (TIAR). (**C**) VM regulation by zinc finger Ran-binding domain-containing protein 2 (ZRANB2). (**D**) VM regulation by insulin-like growth factor 2 mRNA-binding protein 2 (IGF2BP2) (created in BioRender).

**Figure 5 ijms-26-07976-f005:**
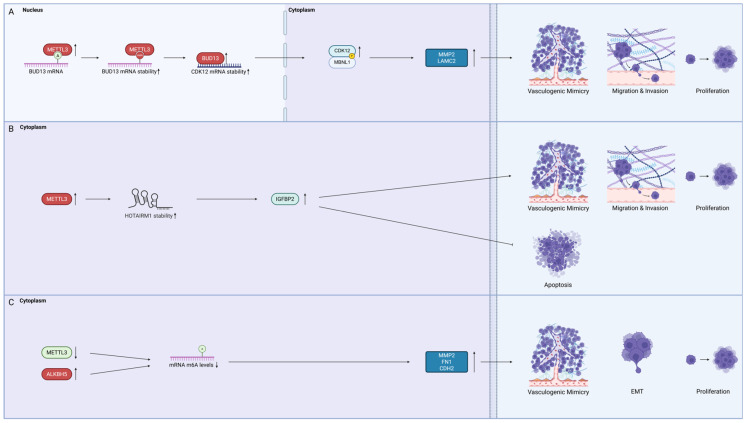
The role of methyltransferase-like 3 (METTL3) remains controversial as there are conflicting reports on its differential expression in GBM. RBPs that promote VM are shown in red, while those that inhibit VM are shown in green. Transcription factors are coloured light blue and other pathway proteins are coloured very light blue. N6-methyladenosine (m^6^A) modifications are coloured red, and phosphorylation is coloured yellow. (**A**) VM regulation pathway mediated by METTL3 through m^6^A modification. (**B**) VM regulation pathway mediated by METTL3 through RNA stabilisation. (**C**) Opposite report on how decrease m^6^A levels led to the increase in GBM VM (created in BioRender).

**Table 1 ijms-26-07976-t001:** Summary of known RBPs controlling GBM VM.

RBP	Expression Level in GBM	Effect of the RBP on GBM VM	Effect of the RBP on Target Transcripts Involved in GBM VM	Cell Lines	Experimental Models Verifying VM Formation	Citation
HuR	Increased	Promoted	↑ *VEGFA* mRNA	U251	3D sphere sprouting assay, tube formation assay	[68]
PUM2	Decreased	Suppressed	↓ *CEBPD* mRNA	U251, U373	Tube formation assay, nude mouse xenograft	[69]
HNRNP A2/B1	Increased	Promoted	↑ *NFATC3* mRNA	U251, U373	Tube formation assay, nude mouse xenograft	[70]
HNRNP D	Increased	Promoted	↑ *ZHX2* mRNA	U87, U251	Tube formation assay, nude mouse xenograft	[71]
RALY	Increased	Promoted	↑ *FOXD1* mRNA	U251, U373	Tube formation assay, nude mouse xenograft	[72]
RBMS3	Decreased	Suppressed	↓ circHECTD1	U87, U251	Tube formation assay, nude mouse xenograft	[73]
SRSF1	Increased	Promoted	↑ circCMTM3	Patient-derived	Tube formation assay, nude mouse xenograft	[74]
SRSF7	Increased	Promoted	↑ circPLEKHA5	U251, U373	Tube formation assay, nude mouse xenograft	[75]
PABPC5	Increased	Promoted	↑ HCG15	U87, U251	Tube formation assay, nude mouse xenograft	[76]
TIAR	Decreased	Suppressed	↓ LOXL1-AS1	U87, U251	Tube formation assay, nude mouse xenograft	[77]
ZRANB2	Increased	Promoted	↑ SNHG20	U87, U251	Tube formation assay, nude mouse xenograft	[78]
IGF2BP2	Increased	Promoted	↑ OIP5-AS1	U87, U251	Tube formation assay, nude mouse xenograft	[79]
METTL3	Uncertain	Suppressed	--	U87	Tube formation assay, nude mouse xenograft	[80]
		Promoted	↑ HOTAIRM1	U87, U251	Tube formation assay, nude mouse xenograft	[81]
		Promotion	↑ *BUD13* mRNA	U251, U373	Tube formation assay, nude mouse xenograft	[82]

Note: the arrows indicate the up-regulation (↑) or down-regulation (↓), respectively.

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
