# Peer review of "The Roles of RNA-Binding Proteins in Vasculogenic Mimicry Regulation in Glioblastoma"

_ijms, 2025, doi:10.3390/ijms26167976_

Round 1

Reviewer 1 Report

Comments and Suggestions for Authors

  • The article is well written and makes important contributions to the pathology of glioblastomas.

  • This is a valuable review on the role of RNA-binding proteins in VM of glioblastoma. It compiles recent findings with sound structure but requires improvements in mechanistic synthesis, clinical contextualization, and therapeutic evaluation.
  • However, there are a few comments that should be addressed-
  • …bevacizumab demonstrated significant efficacy 78 in prolonging OS for metastatic colorectal cancer patients [18] – seems irrelevant for the aims and scope of the current review
  • 1. Anti-angiogenic Therapy Resistance: “It should be noted that…” - avoid conversational tone

  • Figures 1 and 4 don’t have in-text citations

  • 2. Radiotherapy and Chemotherapy Resistance: “Future studies may focus on elucidating the detailed pathways through which VM induces treatment resistance by a combination of commercially available and patient-derived cells. Additionally, research may also examine the proportion of VM in the overall GBM vasculature post-AAT. These efforts would provide a more comprehensive understanding of the extent to which VM contributes to resistance against combined therapies.” - the paragraph would be better placed in the “Conclusions & Perspectives” section

  • 1.2. Pumilio homolog 2 (PUM2): “Further research may be needed to clarify the roles of PUM2 before considering it as a therapeutic target.” – the statement would be better placed in the “Conclusions & Perspectives” section

  • 2.1. RNA Binding Motif Single Stranded Interacting Protein 3 (RBMS3): “Future studies may focus on how RBMS3 interacts with other pathways to further establish its role as a tumour suppressor in GBM.” - the statement would be better placed in the “Conclusions & Perspectives” section

  • 3.1. Poly(A) binding protein cytoplasmic 5 (PABPC5): “Given the differential roles of PABPC1 and PABPC5 in GBM, further research may be necessary to elucidate their roles in various GBM cell lines and the significance of targeting PABPC5 in GBM.” - the statement would be better placed in the “Conclusions & Perspectives” section

  • 3.4. Insulin-like growth factor 2 mRNA-binding protein 2 (IGF2BP2): “Future research may further explore 363 how m6A recognition contributes to VM formation in GBM.” - the statement would be better placed in the “Conclusions & Perspectives” section

  • 5. Methyltransferase-like 3 (METTL3): “Further research is needed to clarify the roles of METTL3 and m6A methylation in a range of cancer cell lines before considering it as a therapeutic target.” - the statement would be better placed in the “Conclusions & Perspectives” section

  • Exploring the Possibilities of RNA-Binding Proteins as Therapeutic Targets for Vasculogenic Mimicry Formation of Glioblastoma: “Given that the targeting of HuR by juglone can suppress VM in GBM, future studies may investigate whether these HuR inhibitors could also suppress VM in GBM.” - the statement would be better placed in the “Conclusions & Perspectives” section

  • Exploring the Possibilities of RNA-Binding Proteins as Therapeutic Targets for Vasculogenic Mimicry Formation of Glioblastoma: “However, further investigation is required to determine the blood-brain barrier penetration efficiency as well as the dosage limit for clinical applications.” - the statement would be better placed in the “Conclusions & Perspectives” section

  • Exploring the Possibilities of RNA-Binding Proteins as Therapeutic Targets for Vasculogenic Mimicry Formation of Glioblastoma: “Nonetheless, the door to targeting VM 457 formation in GBM through RBPs remains open, and future studies may first focus on val-458 idating the effects of current RBP-inhibiting compounds on VM formation of GBM, before 459 identifying other new inhibitors of RBP.” - the statement would be better placed in the “Conclusions & Perspectives” section

  • Section could benefit from being reorganized: e.g., 5.1 validated compounds, 5.2 experimental inhibitors, 5.3 decoy strategies etc.

  • Refferences that seem unrelated: 12, 13, 18, 53, 54, 63, 88, 113

Refferences  that may be of interest-

Signal transduction molecule patterns indicating potential glioblastoma therapy approaches.

.Onco Targets Ther. 2013

Reviewer 2 Report

Comments and Suggestions for Authors

This is a well-written and comprehensive review focusing on the critical role of RNA-binding proteins (RBPs) in regulating vasculogenic mimicry (VM) in glioblastoma (GBM). The manuscript addresses a timely and underexplored aspect of GBM pathology, with a clear emphasis on translational relevance. The organization is logical, and the figures and summary tables are helpful for comprehension. However, a few sections would benefit from further clarification, additional citations, and critical discussion. Specific suggestions are provided below to improve the overall quality and scientific rigor of the review.

  1. Emphasize how this review differs from existing literature on RBPs or VM in gliomas.
  2. Better connect RBP regulation to canonical VM pathways (e.g., PI3K/Akt, VE-cadherin) through more integrative discussion or diagrams.
  3. Improve figure resolution and expand legends. Define color schemes and include full protein names at least once.
  4. Expand on strategies to overcome blood–brain barrier limitations in delivering RBP-targeted therapies.
  5. Briefly discuss how GBM heterogeneity may explain the conflicting roles of METTL3.
  1. Table 1 is informative. Consider adding a new column to indicate the experimental model used (e.g., cell line, mouse model, patient-derived xenograft) to provide additional context.
  2. Include recent clinical trial information or database tables relevant to RBP-targeted agents in glioma or brain tumors broadly.
  3. Expand the perspectives section to include possible AI/omics-based discovery pipelines for RBP regulators in GBM.

Finally, the manuscript is a valuable and timely contribution. With these revisions to enhance clarity, visual communication, and mechanistic integration, it will be suitable for publication in IJMS.

Reviewer 3 Report

Comments and Suggestions for Authors

The manuscript by Pok Kong Tsoi et al. offers a comprehensive and timely overview of the emerging role of RNA-binding proteins (RBPs) in the regulation of vasculogenic mimicry in glioblastoma—a process increasingly recognized as a key mechanism underlying resistance to conventional therapies, including anti-angiogenic treatment, chemotherapy, and radiotherapy. By systematically exploring the involvement of RBPs in vasculogenic mimicry, this review addresses a significant gap in the current literature and provides a level of synthesis and mechanistic detail that has been largely lacking in previous works.

I have some requests to improve the manuscript:

To further enhance the manuscript, I suggest the authors expand their discussion on the heterogeneity of glioblastoma cell lines by providing a more detailed analysis of how RBP expression patterns and vasculogenic mimicry-related signaling pathways differ across molecular subtypes (e.g., proneural versus mesenchymal) and according to IDH mutation status.

Some figures need improvement of resolution such as figure 3 and 4.

The authors are encouraged to introduce their discussion on the role of heparanase inhibition in glioblastoma biology and vascular remodeling. Heparanase  could act upstream of or in parallel with the other pathways, and its inhibition may indirectly modulate the function of the same RNA-binding proteins involved in vasculogenic mimicry. In particular, two studies may offer valuable mechanistic insights relevant to the context of vasculogenic mimicry . One study demonstrated that a novel heparanase inhibitor (RDS3337) effectively suppressed tissue factor overexpression and NF-κB pathway activation in platelets and endothelial cells stimulated by anti-β2-glycoprotein I antibodies (Journal of Thrombosis and Haemostasis 2021). These findings underscore the broader role of heparanase in driving proinflammatory and procoagulant responses, mechanisms that may contribute to vasculogenic mimicry-associated vascular abnormalities in glioblastoma.

A second study showed that treatment with RDS3337 in U87 glioblastoma cells disrupted autophagic flux and promoted apoptosis, indicating a shift in the balance between autophagy and cell death (Cells 2023). This is particularly relevant given the manuscript's discussion of heparanase-regulated pathways—such as PI3K/Akt and MMPs—in vasculogenic mimicry formation. Integrating these findings would enhance the mechanistic depth of the review and further support the rationale for targeting heparanase as a potential therapeutic strategy in glioblastoma, particularly in the context of vasculogenic mimicry regulation and tumor cell fate modulation.

Round 2

Reviewer 3 Report

Comments and Suggestions for Authors

The authors responded to my requests and in my opinion the manuscript
deserves to be published.